# Predictive significance of joint plasma fibrinogen and urinary alpha-1 microglobulin-creatinine ratio in patients with diabetic kidney disease

**Lianlian Pan**[1], **Mingyi Wo**[2], **Chan Xu**[3], **Yan Wu**[4], **Yali Ye**[1], **Fan Han**[2], **Xianming Fei**[2]*, **Fengjiao Zhu**[1]*

**1** Department of Laboratory Medicine, Sanmen People's Hospital, Sanmen, Zhejiang, China, **2** Department of Clinical Laboratory, Laboratory Medicine Center, Zhejiang Provincial People's Hospital (Affiliated People's Hospital, Hangzhou Medical College), Hangzhou, Zhejiang, China, **3** Department of Laboratory Medicine, Affiliated Third Hospital of Zhejiang Traditional Chinese Medicine University, Hangzhou, Zhejiang, China, **4** Department of Laboratory Medicine, Lin'an First People's Hospital, Hangzhou, Zhejiang, China

* sm.zfj@126.com (FZ); feixianming@hmc.edu.cn (XF)

**Data Availability Statement:** Data cannot be shared publicly because the data used to support the findings of this study were from the specific T2D patients, and contain some potentially

## Abstract

### Background

Although many biomarkers have high diagnostic and predictive power for diabetic kidney disease (DKD), less studies were performed for the predictive assessment in DKD and its progression with combined blood and urinary biomarkers. This study aims to explore the predictive significance of joint plasma fibrinogen (FIB) concentration and urinary alpha-1 microglobulin-creatinine (α1-MG/CR) ratio in DKD.

### Methods

A total of 234 patients with type 2 diabetes were enrolled, and their clinical and laboratory data were retrospectively assessed. A ROC curve analysis was performed to evaluate the power of plasma FIB and urinary α1-MG/CR ratio for identifying DKD and advanced DKD, respectively. The predictive power for DKD and advanced DKD was analyzed by regression analysis.

### Results

Plasma FIB and urinary α1-MG/CR levels were higher in patients with DKD than with pure T2D (p<0.001). The multivariate-adjusted odds ratios (ORs) were 5.047 (95%CI: 2.276–10.720) and 2.192 (95%CI: 1.539–3.122) (p<0.001) for FIB and α1-MG/CR as continuous variables for DKD prediction, respectively. The optimal cut-off values were 3.21 g/L and 2.11mg/mmol for identifying DKD, and 5.58 g/L and 11.07 mg/mmol for advanced DKD from ROC curves. At these cut-off values, the sensitivity and specificity of joint FIB and α1-MG/CR were 0.95 and 0.92 for identifying DKD, and 0.62 and 0.67 for identifying advanced DKD, respectively. The area under curve was 0.972 (95%CI: 0.948–0.995) (p<0.001) and 0.611, 95%CI: 0.488–0.734) (p>0.05). The multivariate-adjusted ORs for joint FIB and

identifying or sensitive patient information. Data are available from the Department of Research of Zhejiang Provicial People's Hospital (contact via Tel: 086-0571-85893677, or Email: kjk3929@163.com) for researchers who meet the criteria for access to confidential data.

**Funding:** This work was supported by the Science and Technology Project of Sanmen, Zhejiang, China [Grant number 20305] to Dr. Lianlian Pan, the Medicine and Health Science and Technology Project of Zhejiang Province, China (Grant number 2020KY022 and 2021KY060) to Dr. Xianming Fei (https://wsjkw.zj.gov.cn/), and the Zhejiang Province Public Welfare Technology Application Research Project, China (Grant number LGD21H020004) to Dr. Xianming Fei (https://kjt.zj.gov.cn/). The funders had no role in study design, data collection and analysis, decision to publish, or preparation of the manuscript.

**Competing interests:** The authors have declared that no competing interests exist.

α1-MG/CR at the cut-off values were 214.500 (95%CI: 58.054–792.536) and 3.252 (95%CI: 1.040–10.175) (p<0.05), respectively.

## Conclusion

The present study suggests that joint plasma FIB concentration and urinary α1-MG/CR ratio can be used as a powerful predictor for general DKD, but it is less predictive for advanced DKD.

## Introduction

Diabetes is one of the biggest health care challenges of the 21st century, and affects >422 million people [1], and seriously threatens to human health worldwide. The microvascular complications are the common and serious complications of diabetes [2]. In these complications, diabetic kidney disease (DKD), also named as diabetic nephropathy (DN), is the common one, which is the leading cause of chronic kidney disease [3]. DKD typically manifests a progressive deterioration in kidney function such as augmented glomerular filtration rate (GFR), glomerular hypertrophy, and urinary leakage of albumin [4], and is associated with poor outcomes of patients [5], and is a predictor of mortality in diabetes [6]. Therefore, early identifying and diagnosis of DKD could provide the opportunity to intervene the development and progression of DKD, which would be more important to decrease the morbidity and mortality of DKD, and further to improve the prognosis of diabetic patients.

DKD is pathologically graded four stages from stage I to IV based on the condition of renal imparement according to the Renal Pathology Society classification [7], and the stage 4 represents the progression to end-stage renal disease [8]. In clinical practice, urinary albumin excretion rate is a good marker popularly recognized for diagnosis and progression of DKD. However, microalbumin urine is not a sensitive and specific predictor for prevalence and progression of DKD, as the sensitivity and specificity of microalbuminuria for early detection of disease are limited by a number of factors: high day-to-day variability of urine albumin excretion; the phenomenon of non-albuminuric diabetic nephropathy; and presence of advanced renal pathological changes by the time microalbuminuria is clinically detectable [9]. Although some studies have revealed that many biological markers or biomarkers can reflect the presence of microvascular damage in type 2 diabetes (T2D) patients [10, 11], and are associated with development of T2D [12, 13], their predictive power for DKD and its progression is still limited.

It has been well recognized that many biomarkers play potential roles in incidence and prevalence of diabetic kidney disease [14, 15]. Although proteinuria is a good clinical indicator of early kidney injure, various urine proteins may demonstrate different dignositic and predictive power. However, their values of diagnosis and prediction for DKD and its progression did not well revealed in T2D patients. Our previous studies have indicated that plasma fibrinogen [16], and blood monocyte-lymphocyte ratio [17], can be used as predictors of diabetic nephropathy, which suggests that blood biomarkers may be good predictors for discriminating between pure T2D and DKD. However, whether the combinations of plasma fibrinogen and some urinary proteins are more effective for risk prediction and more powerful in identification of general DKD or advanced DKD, respectively, has not been properly studies with DKD patients diagnosed pathologically. Therefore, the main purpose of this study was to explore the

clinical predictive significance of plasma fibrinogen combined urinary proteins for DKD and its progression in T2D patients.

## Materials and methods

### Patients population

This study was a retrospective cross-sectional investigation. In this study, we randomly assessed the T2D patients from the Departments of Endocrinology, Zhejiang Provincial People's Hospital, China, between 2015 and 2021, and the same numbers of patients with pure T2D (control group) and DKD (case group) were included. All patients were diagnosed according to the criteria in diabetes guideline of China Diabetes Association [18–20]. Before treatment, the disease history, traditional risk factors, physical examination, physiological data and laboratory parameters of all T2D patients from hospital information system were collected. The major inclusion criteria included: 1) type 2 diabetes without microvascular complications diagnosed clinically; 2) type 2 diabetes with kidney disease confirmed by renal puncture pathology; 3) kidney disease before treatment. The major exclusion criteria were as follows at least: 1) other diabetic complications except for microvascular complications; 2) primary liver and kidney dysfunctions; 3) post-operation; 4) cardiovascular and cerebrovascular diseases; 5) malignancies; 6) acute inflammation and infections. The DKD patients were assigned two groups including stage II+III (non-advanced DKD) and stage IV (advanced DKD) according to the pathological diagnosis. Because this study was a retrospective review on the data of discharged patients without using their privacy, and would not report their detailed information, thus the participant consent was waived by the ethical committee of the hospital. The study was approved by the Ethical Committee of Zhejiang Provincial People's Hospital (Approval No.: 2021KT014, Date of Registration: June 21, 2021).

### Major parameters assay

The study reviewed the major sera biochemical parameters, plasma fibrinogen and D-dimer, urinary proteins and other biomarkers. Brief assay methods of all reviewed laboratory parameters were as follows: 1) Sampes collection and treatment: venous blood samples were collected in sodium citrate- or EDTA-$K_2$-containing and anticoagulant-free vaccutainer tubes (Becton Dickinson, MountainView, CA, USA) after an overnight fast before treatment. Sera and plasma were obtained by centrifugation at 1500$g$ at room temperature for 10 minutes. 2) Sample detection: The major biochemical indexes in sera and urinary proteins (including microalbumin and globulins) concentrations, creatinine, and other biochemical indicators in spot urine were measured by a biochemical analyzer (AU5800, Beckman-coulter company, USA). Plasma fibrinogen and d-dimer were also measured by an coagulation analyzer and commercially available reagents (CS-5100, Sysmex Inc., Japan). The estimated glomerular filtration rate (eGFR) was calculated based on the formula: eGFR = 133× $(CysC/0.8)^{-0.449\ (-1.328\ if\ CysC>0.8mg/L)}$ × $0.996^{age}$ × (0.932 if female) [21]. The whole urine of the patients during 24h was collected for the measurement of urinary albumin excretion (UAE). At the same time, the biological and clinical data (age, gender, body mass index, and smoking status for all subjects, and T2D-related data before treatment, which might be confounders of the predictors) were gathered and reviewed.

### Statistical analysis

In this study, although some parameters had a few missing data, it did not significantly influence the results of the study because there was enough samples included. For all parameters, data were first tested for distribution normality by the Kolmogorov-Smirnov test, and

normally and non-normally distributed data were presented as mean±standard deviation and median, respectively. For samples of non-normal distribution data and normal distribution data, Mann-Whitney *U*-test and Student's *t*-test were used in DKD patients with different clinical characteristics, respectively. Categorical data (percentage) were analyzed by Chi-square test. Receiver operating characteristic (ROC) curve was constructed, and the area under the curve was calculated to evaluate the power including sensitivity and specificity for discriminating between pure T2D and DKD as well as different stages of DKD. Regression analysis was performed to calculate the odds ratio (OR) and its 95% confidence interval for general DKD and advanced DKD. Statistical analyses were performed using statistical package SPSS 20.0 (SPSS, Chicago Illinois, USA). *P* value of less than 0.05 was considered statistically significant.

## Results

### Basic characteristics and biomarkers of pure T2D and DKD patients

In this study, the associations of all parameters levels with pure T2D and DKD patients were evaluated. A total of 234 matched patients were reviewed, and the subjects included 169 males and 65 females aged 35–76 years, 117 patients with pure T2D and 117 ones with DKD. In 117 DKD patients, no subjects were in stage I, because most of these patients do not demonstrate proteinuria (normoalbuminuria), and physicians might not consider to perform a renal puncture. There were 81 non-DKD patients including 46 patients in stage II (27 in IIa, 19 in IIb) and 35 in stage III, and 36 advanced DKD patients (in stage IV). In their basic characteristics, most parameters except BMI, males and smoking percentage showed statistical difference between pure T2D and DKD (p<0.05). In measured biomarkers, only GLU levels were not statistically significant (p<0.05) between the two groups. The results exhibited that the patients with DKD were more likely to have abnormal levels of blood biomarkers and urinary proteins than that of T2D without complications. The data were presented in Table 1. In the pure T2D and DKD patients, there was some missing results to different extent except for the Sex and Age, and the missing percent of variables was from 1.70% to 56.4%, which urinary biomarkers had higher missing percent than others because some of them were not routine indicators for observation. Thus, the patients with missing results would be excluded in the following statistical analysis of relative indicators.

### Incidence and risk factor analysis of DKD

In the study, we further divided the 117 DKD patients into two groups including test group (59 subjects) and validation group (58 subjects) for subsequent study. For all the indicators, there was no remarkable differences between the two groups (p>0.05). Data were not presented. Excluding the patients with missing results, we further observed the incidence of DKD, and also evaluated the risk factors for DKD by univariate regression analysis in test group. The observed variables included all basic characteristics and those with P-value of less than 0.05. When the cut-off points of continuous variables were set at the median, the incidence of DKD demonstrated significant differences for all variables except for BMI (p<0.05); Moreover, the majority of the parameters when being continuous variable showed an OR value with statistical significance in prediction for DKD. Detailed results were in Table 2.

### Adjusted-multivariate regression analysis of risk factors for DKD

Based the above univariate analysis, we analyzed all the biological and pathological characteristics of patients by multivariate regression analysis, and the results showed that increased age, high SBP and DBP, and long duration of diabetes were the independent risk factors for DKD

**Table 1. Comparisons of clinical and laboratory characteristics of pure T2D and DKD patients.**

| Parameters | Pure T2D | | DKD | | Statistical value | P-value[a] |
|---|---|---|---|---|---|---|
| | n | value | n | value | | |
| Sex, male (n, %) | 117 | 88, 75.2 | 117 | 82, 70.1 | 1.044 | 0.307 |
| Age (yr) | 117 | 38.4±11.1 | 117 | 54.1±10.6 | 11.044 | <0.001 |
| SBP (mmHg) | 117 | 127.4±15.3 | 113 | 146.0±22.8 | 7.768 | <0.001 |
| DBP (mmHg) | 117 | 78.2±10.0 | 112 | 81.4±12.3 | 2.170 | 0.031 |
| Duration of disease (yr) | 106 | 2.0 (1–15) | 107 | 10.0 (0.1–30.0) | 1805.5 | <0.001 |
| BMI (kg/m$^2$) | 117 | 25.54 (17.53–46.46) | 105 | 24.8 (19.5–57.2) | 6124.000 | 0.969 |
| Smoking (yes, n, %) | 107 | 27, 25.2 | 117 | 27, 23.1 | 0.142 | 0.706 |
| HbA1c (%) | 117 | 9.33±2.73 | 104 | 7.55±1.97 | 4.577 | <0.001 |
| GLU (mmol/L) | 117 | 6.46 (2.44–25.16) | 115 | 6.28 (0.81–22.1) | 6021.000 | 0.167 |
| FIB (g/L) | 83 | 2.35 (1.48–5.22) | 103 | 3.84 (1.40–9.36) | 991.000 | <0.001 |
| D-D (ng/ml) | 82 | 150.0 (40.0–23870.0) | 98 | 700.0 (30–8410) | 1024.000 | <0.001 |
| eGFR (ml/24h) | 110 | 110.10 (49.90–181.51) | 66 | 43.76 (8.62–119.0) | 274.000 | <0.001 |
| α1-MG/CR (mg/mmol) | 97 | 0.70 (0.12–15.06) | 90 | 9.075 (1–105.0) | 242.000 | <0.001 |
| ACR (mg/g) | 117 | 0.94 (0.01–0.73) | 91 | 273.7 (1.14–1098.4) | 129.000 | <0.001 |
| TRF/CR (mg/mmol) | 68 | 0.080 (0.010–0.73) | 88 | 17.59 (0.15–461.50) | 22.500 | <0.001 |
| IgG/CR (mg/mmol) | 74 | 4.00 (0.17–23.95) | 90 | 36.05 (0.64–413.30) | 707.000 | <0.001 |

DATA are presented as mean±standard deviation, median (minimum-max value) or percentage. DKD, diabetic kidney disease; BMI, body mass index; SBP, Systolic blood pressure; DBP, diastolic blood pressure; HbA1c, glycosylated hemoglobin; GLU, serum glucose; FIB, fibrinogen; D-D, d-dimer; eGFR, estimated glomerular filtration rate; α1-MG, α1-microglobulin; ACR, albumin-creatinine ratio; TRF, transferrin; IgG, immunoglobulin G; CR, creatinine.

[a]P values were calculated by student's *t*-test, Mann-Whitney *U*-test and *Chi-square* test, respectively.

(analyzed results were not presented). Therefore, we set the five characteristics also including the sex as adjusted-factors, then conducted adjusted-multivariate analysis for other risk factors (ACR and eGFR were not included because they are traditional diagnostic biomarkers of DKD), and screened out three independent risk factors for DKD, including FIB (OR: 5.047, 95%CI: 2.276–10.720, p<0.001), α1-MG/CR (OR: 2.334, 95%CI: 1.586–3.533, p<0.001), and IgG/CR (OR:1.193, 95%CI: 1.015–1.403, p<0.05).

## ROC curve analysis for identifying DKD

In the independent risk factors in test group, besides plasma FIB, we selected α1-MG/CR as the focus variable because it had a higher OR value than that of IgG/CR. Subsequently, we first treated FIB and α1-MG/CR as the combined prediction probability (combined PRE), and further constructed a ROC curve of FIB and α1-MG as well as combined PRE for identifying DKD from pure T2D, respectively, and obtained their optimal cut-off values. From the ROC curves, the area under curve (AUC) was 0.858, 0.969 and 0.972, and the optimal cut-off values were 3.17 g/L, 2.65 mg/mmol, and 0.19 for FIB, α1-MG/CR, and combined PRE, respectively. Based on the optimal cut-off values, plasma FIB had a relatively low sensitivity (0.73) and a high specificity (0.89), but urinary α1-MG/CR showed a high sensitivity and specificity of 0.93 and 0.95, respectively. When FIB combined α1-MG/CR, a high specificity (0.92) and an increased sensitivity (0.97) was observed. In order to validate the results, we further used the above cut-off values in validation group, and the calculated sensitivity and specificity were 0.80 and 0.89 for FIB, 0.85 and 0.95 for α1-MG/CR, and 0.92 and 0.94 for joint FIB and α1-MG/CR, which was similar to that in test group (p>0.05 for all). Finally, we included all T2D patients to construct ROC curves. The results also demonstrated that FIB, α1-MG/CR, and

**Table 2. Incidence and univariate analysis of DKD in different characteristics of T2D patients.**

| Comparison | Prevalence of DKD (%) | OR (95%CI) | P-value[a] |
|---|---|---|---|
| Sex, male vs. female | 32 vs. 38 | 0.761 (0.375–1.504) | >0.05 |
| Age (years), ≥41 vs. <41 | 53 vs .12 | 1.103 (1.065–1.141) | <0.001 |
| Duration of diabetes (years), ≥3 vs. <3 | 49 vs. 12 | 1.311 (1.198–1.435) | <0.001 |
| Smoking habit, yes vs. no | 33 vs. 37 | 0.837 (0.394–1.781) | >0.05 |
| BMI (kg/m$^2$), ≥25.45 vs. <25.45 | 30 vs. 31 | 1.028 (0.963–1.097) | >0.05 |
| SBP (mmHg), ≥130 vs. <130 | 48 vs. 16 | 1.054 (1.032–1.076) | <0.001 |
| DBP (mmHg), ≥78 vs. <78 | 38 vs. 25 | 1.035 (1.004–1.067) | <0.05 |
| HbA1c (%), ≥8.5 vs. <8.5 | 18 vs. 45 | 0.704 (0.596–0.832) | <0.001 |
| FIB (g/L), ≥ 2.70 vs. < 2.70 | 64 vs. 12 | 5.676 (3.086–10.439) | <0.01 |
| D-D (ng/ml), ≥205 vs. <205 | 65 vs. 9 | 1.000 (1.000–1.001) | <0.05 |
| eGFR (ml/24h), ≥104.7 vs. <104.7 | 1 vs. 45 | 0.894 (0857–0.932) | <0.001 |
| α1MG/CR (mg/mmol), ≥1.02 vs.< 1.02 | 60 vs. 1 | 2.144 (1.602–2.867) | <0.001 |
| ACR (mg/g), ≥1.5 vs. <1.5 | 51 vs. 1 | 4.726 (1.212–18.426) | <0.05 |
| TRF/CR (mg/mmol), ≥0.16 vs. <0.16 | 75 vs. 0 | 2.935 (1.129–16.271) | <0.01 |
| IgG/CR (mg/mmol), ≥4.0 vs. <4.0 | 46 vs. 19 | 1.191 (1.093–1.297) | <0.001 |

The cut-off points of continuous variables were the median values, and OR values were obtained from univariate regression analysis based on the quantitative indicators as continuous variables.

DKD, diabetic kidney disease; T2D, type 2 diabetes; BMI, body mass index; SBP, Systolic blood pressure; DBP, diastolic blood pressure; HbA1c, glycosylated hemoglobin; GLU, serum glucose; FIB, fibrinogen; D-D, d-dimer; eGFR, estimated glomerular filtration rate; α1-MG, α1-microglobulin; ACR, albumin-creatinine ratio; TRF, transferrin; IgG, immunoglobulin G; CR, creatinine. OR, odds ratio; CI, confidence interval.

[a]P values, were calculated by univariate regression analysis.

combined PRE had high AUC of 0.884 (95%CI: 0.834–0.934), 0.972 (95%CI: 0.952–0.993), and 0.972 (95%CI: 0.948–0.995) (p<0.05), respectively; and they also exhibited high sensitivity and specificity based on their optimal cut-off values (Fig 1). For following study, we divided the patients into two groups according to the cut-off values, respectively.

## Adjusted-multivariate regression analysis of FIB and α1-MG/CR for DKD

Based on the ROC curve analysis, FIB and α1-MG/CR were treated as categorical variables by the optimal cut-off values of 3.17g/L and 2.65 mg/mmol in test group, respectively. The adjusted-multivariate regression analysis revealed that FIB and α1-MG/CR had an OR of 32.751 (95%CI: 7.927–135.322) and 27.943 (95%CI: 6.921–130.892). respectively, when as categorical variables for DKD risk. In validation group, when FIB and α1-MG/CR were treated as categorical variables by the cut-off values of 3.17g/L and 2.65 mg/mmol, each also showed a high OR value for DKD prediction. Finally, by using the cut-off values of 3.215 g/L, 2.11 mg/mmol and 0.25 from the ROC analysis in all T2D patients for discriminating between pure T2D and DKD, we also performed the adjusted-multivariate regression analysis. In the study, each of them exhibited a high hazards ratio, the OR value was 49.939 (13.171–189.340), 34.634 (8.756–136.995), and 214.500 (58.054–792.536), (p<0.05), respectively. Data were in Table 3.

## Correlation analysis of plasma FIB with urinary proteins

In order to observe the associations of plasma FIB with urinary proteins, the correlation of FIB concentration with α1-MG/CR and IgG/CR, two independent risk factors with higher ORs in urinary proteins for DKD, was analyzed, respectively. However, there was no significant

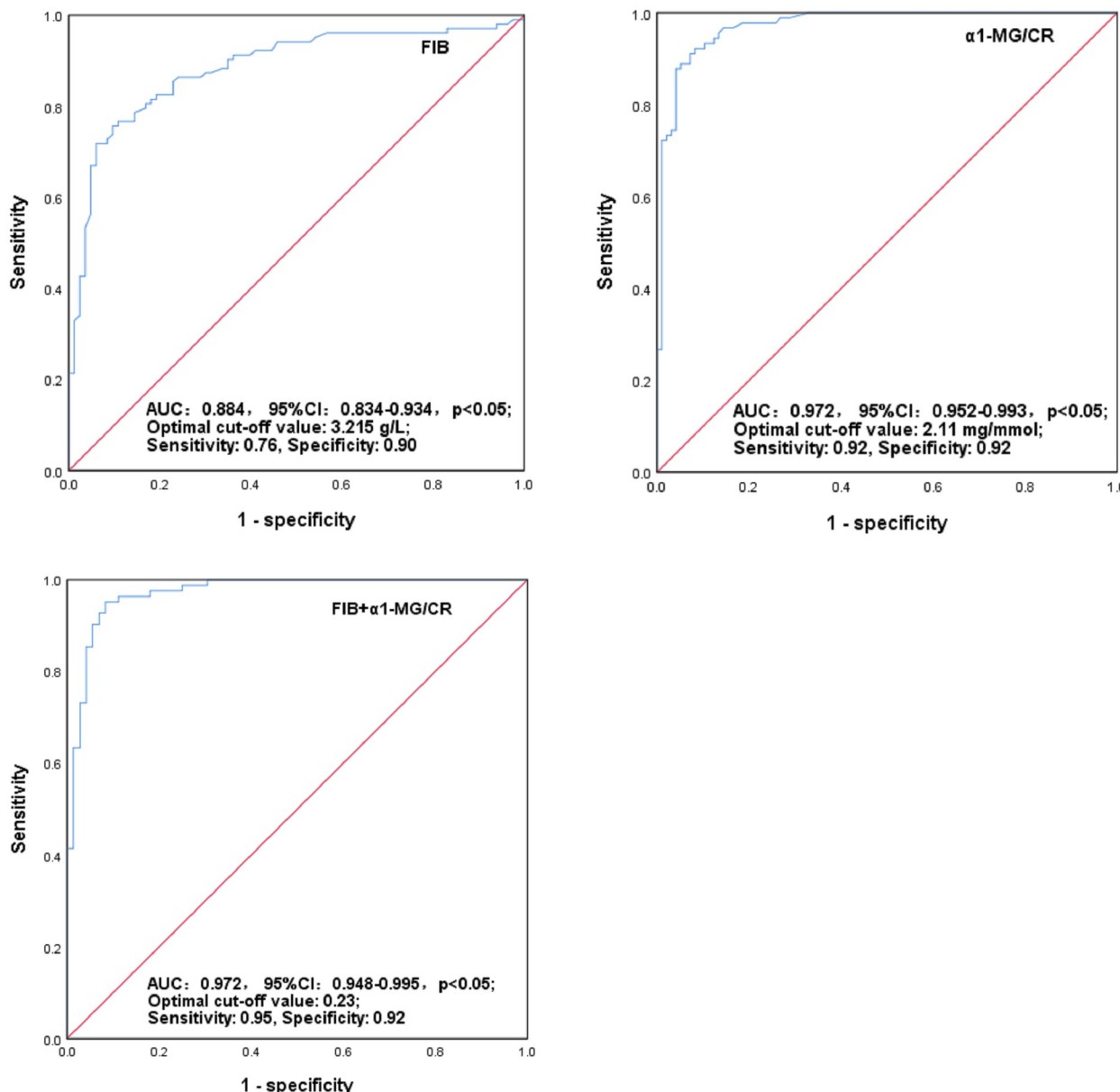

**Fig 1. ROC curves of joint FIB and α1-MG/CR in identifying patients with DKD.** Curve for joint FIB and α1-MG/CR was constructed by using the prediction probability. DKD: diabetic kidney disease; T2D: type 2 diabetes; FIB: fibrinogen; MG: microglobulin; CR: creatinine; ROC: receiver operating characteristic. AUC: area under curve.

correlation between their levels (p>0.05). For assessing the associations of FIB and α1-MG/CR with DKD progression, we further divided all DKD patients into three groups of stage II, III and IV, respectively, according to the pathological diagnosis. The results indicated that levels of all indicators did not have significant difference among the three groups (p>0.05 for all). Subsequently, when groups stage II+III and stage IV were compared, also no significant difference was found (p>0.05 for all). The comparisons of different groups for plasma FIB and urinary α1-MG/CR were as follows: $F$ = 0.996 and 0.352 by ANOVA among three groups, $U$ = 983.500 and 688.500 by Mann–Whitney $U$ test between stage II+III and stage IV (p>0.05 for all), respectively.

**Table 3. Adjusted multivariate analysis of joint FIB and a1-MG/CR for DKD risk.**

| Variables | | | OR (95%CI) | P-value[a] |
|---|---|---|---|---|
| **Test group** | FIB | Continuous variable | 5.047 (2.276–10.720) | <0.001 |
| | | Optimal cut-off from ROC curve | | |
| | | <3.17 g/L | 1.000 | |
| | | ≥3.17 g/L | 50.799 (10.777–239.400) | <0.001 |
| | α1-MG/CR | Continuous variable | 2.192 (1.539–3.122) | <0.001 |
| | | Optimal cut-off from ROC curve | | |
| | | <2.65 mg/mmol | 1.000 | |
| | | ≥2.65 mg/mmol | 34.945 (6.061–136.203) | <0.001 |
| **Validation group** | FIB | Continuous variable | 7.221 (2.761–18.886) | <0.001 |
| | | Optimal cut-off from ROC curve | | |
| | | <3.17 g/L | 1.000 | |
| | | ≥3.17 g/L | 54.006 (9.112–320.087) | <0.001 |
| | α1-MG/CR | Continuous variable | 1.573 (1.232–2.008) | <0.001 |
| | | Optimal cut-off from ROC curve | | |
| | | <2.65 mg/mmol | 1.000 | |
| | | ≥2.65 mg/mmol | 27.993 (5.781–135.554) | <0.001 |
| **Total group** | FIB | Continuous variable | 5.999 (2.912–12.360) | <0.001 |
| | | Optimal cut-off from ROC curve | | |
| | | <3.215 g/L | 1.000 | |
| | | ≥3.215 g/L | 49.939 (13.171–189.340) | <0.001 |
| | α1-MG/CR | Continuous variable | 2.273 (1.483–3.484) | <0.001 |
| | | Optimal cut-off from ROC curve | | |
| | | <2.11 mg/mmol | 1.000 | |
| | | ≥2.11 mg/mmol | 34.634 (8.756–136.995) | <0.001 |
| | FIB+α1-MG/CR | Continuous variable | 23.264 (4.871–40.287) | <0.001 |
| | | Optimal cut-off from ROC curve | | |
| | | <0.25 | 1.000 | |
| | | ≥0.25 | 214.500 (58.054–792.536) | <0.001 |

The results were analyzed by adjusting risk factors of sex, age, SBP, DBP and duration of disease when the markers were presented as continuous and categorical variables based on the cut-off values, respectively. The variable of joint FIB and α1-MG/CR was presented as prediction probability.

DKD: diabetic kidney disease; SBP: Systolic blood pressure; DBP: diastolic blood pressure; FIB: fibrinogen; α1-MG: α1-microglobulin; CR: creatinine; OR: odds ratio; CI: confidence interval.

[a]*P* values were calculated by multivariate regression analysis.

### ROC curve and regression analysis for DKD progression

Although there was not remarkable difference between different stages of DKD, we also aimed to assess the predictive power of joint plasma FIB and urinary α1-MG/CR (combined PRE) for advanced DKD by the cut-off values from ROC analysis. We divided the DKD patients into two parts in random, one included stage II/III (non-advanced DKD), and stage IV (advanced DKD) was in another. ROC curve analysis showed a low area under curve of 0.574 (95%CI: 0.446–0.703), 0.558 (95%CI: 0.418–0.698), and 0.611 (95%CI: 0.488–0.734) for FIB, α1-MG/CR, and the combined PRE, respectively, in discriminating between non-advanced and advanced DKD. When the optimal cut-off values were set as 5.58 g/L, 11.07 mg/mmol and 0.37, each had a very low sensitivity (0.33, 0.39, and 0.62, respectively). The FIB, α1-MG/CR, and the combined PRE were further treated as categorical variables by their cut-off values, and the multivariate regression analysis revealed that each variable exhibited a high OR value in

**Table 4. Power of joint FIB and α1-MG/CR in identifying advanced DKD by ROC curves.**

| Variables | AUC (95%CI) | Optimal cut-off value | Sensitivity | Specificity |
|---|---|---|---|---|
| FIB | 0.574 (0.446–0.9703) | 5.58 g/L | 0.33 | 0.91 |
| α1-MG/CR | 0.558 (0.418–0.698) | 11.07 mg/mmol | 0.39 | 0.85 |
| FIB+α1-MG/CR | 0.611 (0.488–0.734) | 0.37 | 0.62 | 0.67 |

Advanced DKD were the stage IV of DKD.

DKD: diabetic kidney disease; FIB: fibrinogen; α1-MG: microglobulin; CR: creatinine; AUC: area under curve; ROC: receiver operating characteristic; CI: confidence interval.

prediction for advanced DKD [FIB (OR: 5.052, 95%CI: 1.525–16.737), α1-MG/CR (OR: 3.332, 95%CI: 1.187–9.355), and FIB+α1-MG/CR (OR: 3.252, 95%CI: 1.040–10.175), p<0.05 for all]. Detailed results were presented in Table 4.

## Discussion

The present study observed the basic characteristics of 234 patients with T2D, and measured some biochemical markers in blood and urine. We also assessed the associations of joint plasma FIB and urinary α1-MG/CR ratio with DKD and advanced DKD. Two of the major findings in the study were that combined plasma FIB and urinary α1-MG/CR had high identifying power for DKD from pure T2D, and that their combination can be used as an independent and powerful predictor for genernal DKD.

As one of diabetic complications, DKD patients may exhibit the more abnormalities of blood and urinary biomarkers than that of pure T2D patients as what the results exhibited in this study, such as high plasma FIB concentration and urinary α1-MG/CR ratio, which was sure to be associated with the metabolic disorders, and was also consistent with what reported by our and other authors' studies [16, 22]. It has been recognized that many indicators including biologic and pathological factors contribute to development and progression of DKD [23, 24], and several factors have been clarified to be related to DKD progression, such as hyperglycemia, hypertension, dyslipidemia, as well as duration of DKD [25]. In this study, plasma FIB and urinary α1-MG/CR not only were associated with increased prevalence of DKD, but also exhibited much higher OR values by univariate analysis than other biomarkers for DKD prediction, respectively. Moreover, through adjustment for potential confounders including age, sex, duration of DKD, and blood pressure, multivariate analysis also demonstrated high ORs, which indicated that increased plasma FIB concentration and urinary α1-MG/CR ratio were independent risk factors probably with high predictive power. Besides the findings in this study, our previous study has partly revealed the association of plasma FIB with DKD [16], and Jiang et al. [26] also indicated that α1-MG/CR was negatively correlated to eGFR, and was a risk factor for decreased eGFR. Therefore, all these analyses suggest that there is definitely close associations of plasma FIB and urinary α1-MG/CR with DKD risk.

In development and progression of T2D and DKD, some basic biomarkers in blood and urine may directly demonstrate the status of disease, and are sure to be good predictors [27–30]. In this study, when using plasma FIB and urinary α1-MG/CR as well as their combination as predictive factors, their optimal cut-off points would be needed to categorize patients with and without the risk. We obtained the optimal cut-off values of 3.17 g/L and 2.65 mmg/mmol for FIB and α1-MG/CR, respectively, from ROC curves in test group, and observed a modest sensitivity (0.73) for FIB, and high sensitivity and specificity of more than 0.92 for α1-MG/CR and joint FIB and α1-MG/CR for identifying DKD from pure T2D. Moreover, a similar power of identifying DKD was further found in another validation group. And when including all

patients, joint FIB and α1-MG/CR also showed a higher power of identifying DKD than FIB or α1-MG/CR. Therefore, these results suggest that joint plasma FIB and urinary α1-MG/CR is a better predictor for DKD than any other biomarker in the present study, and their combination by the optimal cut-off values could be used as a more sensitive and specific marker for identifying DKD in T2D patients. When FIB and α1-MG/CR were treated as categorical variables to represent "with" and "without" DKD risk based on the cut-off values, our findings indicated that both FIB and α1-MG/CR demonstrated a very high OR by adjusted multivariate analysis, respectively, in test, validation and total patients groups for DKD, and joint FIB and α1-MG/CR even showed an extremely high OR (214.500) in total patients group, and the main causes might be the extreme differences of actual concentrations between patients of DKD and pure T2D. In accordance with this results, we have indicated the plasma FIB was a good risk predictor for diabetic nephropathy (DN) in our previous study [16]. Although relatively few studies have assessed α1-MG in diabetes because it is not routinely used to diagnose and evaluate renal impairment in diabetic subjects, Xu *et al.* [31] also indicated that urine α1-MG/CR was an independent risk factor of new-onset renal dysfunction in T2D patients. And our analyse resluts further indicated that joint plasma FIB and urinary α1-MG/CR is a powerful independent predictor for incidence of DKD in T2D patients.

DKD is usually defined clinically by proteinuria occurrence or declined renal function, e.g. reduced glomerular filtration rate (GFR) [32]. It is clear that urinary proteins including albumin and different globulins can better reflect chronic kidney injure to some extent, and spot urine Protein to Creatinine can be used to detect and monitor diabetic kidney disease [33], but these biomarkers have deficiencies in sensitivity and/or specificity to detect kidney injury [34]. Within our knowledge, the potential significance of urinary proteins for the prediction of advanced DKD is unclear. In order to evaluate the associations of plasma FIB change with urinary proteins to creatinine ratio in DKD patients, we first analyzed the correlation of FIB concentration with urinary α1-MG/CR and IgG/CR. The results showed a low correlation between FIB and α1-MG/CR and IgG/CR, respectively, which indicated that plasma FIB change was not consistent with that of urinary proteins, and also implies that FIB or α1-MG/CR may be an independent assessment factor for DKD. Subsequently, plasma FIB and urinary α1-MG/CR levels were observed in different stages of DKD. However, as independent predictors of DKD, FIB and α1-MG/CR did also not demonstrated remarkable difference in different stages. Further ROC curve analysis also showed a low AUC and sensitivity for FIB, α1-MG/CR, and their combination, respectively, in identifying advanced DKD. The above results indicated that joint FIB and α1-MG/CR have a low power in identifying patients with advanced DKD from non-advanced DKD, suggesting the combination has a lower diagnostic value for advanced DKD. By treating FIB and α1-MG/CR of DKD patients as categorical variables based on the optimal cut-off values, the regression analysis showed that joint FIB and α1-MG/CR had a high OR value (3.252) in prediction of advanced DKD, indicating that the joint FIB and α1-MG/CR was an independent predictor. Gao *et al.* [35] used urinary albumin/creatinine ratio, plasma albumin, and other markers to predict renal function decline in patients with type 2 DKD, and revealed a valuable predictive significance; and the study of Ide *et al.* [36] also demonstrated that urinary tubule injury markers including UCR and their combination were significant predictors for the future eGFR decline in patients with type 2 diabetes. These studies exhibited high predictive power of some blood and urinary markers for DKD progression, but our results did not reveal significant differences for FIB, α1-MG/CR and others except for the glomerular biomarkers between different DKD stages, and they did not have high overall predicting power for advanced DKD, which may be related to the differences of the observed indicators and diagnostic criteria. Thus, in this study, a low identifying power would lead a less predictive efficacy of joint FIB and α1-MG/CR in the prediction of advanced DKD, suggesting

that the joint FIB and α1-MG/CR is not an effective and practical predictor for advanced DKD.

Mechanisms which plasma FIB and urinary α1-MG/CR may be associated with DKD and its progression remain to be investigated. Hyperfibrinogenaemia is common in type 2 diabetes, and synthesis of fibrinogen is up-regulated in type 2 diabetic patients with increased urinary albumin excretion, which implies that high FIB concentration is consistent with albuminuria [37]. However, fibrinogen synthesis is also increased even in the absence of micro- or macroalbuminuria in type 2 diabetes [38], suggesting that plasma FIB change is not consistent with urine albumin excretion, and it is probably a more sensitive and powerful predictor for DKD. Therefore, that may be the major mechanism of high predictive power of increased plasma FIB for general and advanced DKD. Alpha-1 microglobulin (α1-MG) is a small molecular weight protein (27 kDa), and passes freely through the glomerular membranes, and about 99% is reabsorbed and catabolized by the proximal tubular cells [39]. However, other proteins such as albumin and immunoglobulins have higher molecular weight than α1-MG, and can not pass through glomeruli in normal conditions, thus, α1-MG is much easier to excrete in urine even with renal tubules failures but without glomeruli failures, which is probably the important causes why urinary α1-MG especially α1-MG/CR is a more powerful predictor for general DKD than other urinary proteins.

This study had some limitations. First, the population probably included some DKD subjects with different microvascular complications being not diagnosed, which could produce some uncertain results, and further influenced the predictive power. Second, the pure T2D patients were defined by clinical diagnosis based on proteinuria occurrence or declined renal function, which might not exclude some DKD with stage I diagnosed by renal puncture. Therefore, it might decrease the diagnostic and predictive power for identifying DKD. Third, the study did not include enough DKD patients diagnosed by pathology, and there were also not enough subjects to validate the result, which might lead an indefinite result. Despite the limitations, our study also revealed that joint plasma FIB concentration and urinary α1-MG/CR ratio has a higher predictive significance in DKD.

## Conclusions

The present study suggests that joint plasma FIB concentration and urinary α1-MG/CR ratio can be used as a powerful predictor for general DKD. However, because this was a retrospective and cross-sectional study, further multiple-center and prospective cohort studies with larger group of patients may exhibit more definitive results to validate the significance of the combination of plasma fibrinogen and urinary α1-MG/CR.

## Acknowledgments

The authors thank all people who supported them in the collection of patients data, and helped them to perform statistical analysis.

## Author Contributions

**Conceptualization:** Lianlian Pan, Mingyi Wo, Xianming Fei, Fengjiao Zhu.

**Data curation:** Mingyi Wo, Fan Han.

**Formal analysis:** Lianlian Pan, Chan Xu.

**Funding acquisition:** Lianlian Pan, Xianming Fei.

**Investigation:** Chan Xu, Yan Wu.

**Methodology:** Chan Xu, Yan Wu, Yali Ye.

**Project administration:** Fengjiao Zhu.

**Resources:** Lianlian Pan, Fengjiao Zhu.

**Software:** Mingyi Wo.

**Supervision:** Xianming Fei, Fengjiao Zhu.

**Validation:** Yali Ye, Fengjiao Zhu.

**Writing – original draft:** Lianlian Pan, Mingyi Wo, Chan Xu, Yan Wu, Yali Ye, Xianming Fei, Fengjiao Zhu.

**Writing – review & editing:** Xianming Fei, Fengjiao Zhu.

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
