## [Decision Letter · Decision Letter 0]

19 May 2022

PONE-D-22-06480Predictive Significance of Joint Plasma Fibrinogen and Urinary Alpha-1 Microglobulin-Creatinine Ratio in Patients with Diabetic Kidney DiseasePLOS ONE

Dear Dr. Pan, 

Thank you for submitting your manuscript to PLOS ONE. After careful consideration, we feel that it has merit but does not fully meet PLOS ONE’s publication criteria as it currently stands. Therefore, we invite you to submit a revised version of the manuscript that addresses the points raised during the review process.

We look forward to receiving your revised manuscript.

Kind regards,

Donovan Anthony McGrowder, PhD., MA., MSc

Academic Editor

PLOS ONE

Journal Requirements:

Whilst you may use any professional scientific editing service of your choice, PLOS has partnered with both American Journal Experts (AJE) and Editage to provide discounted services to PLOS authors. Both organizations have experience helping authors meet PLOS guidelines and can provide language editing, translation, manuscript formatting, and figure formatting to ensure your manuscript meets our submission guidelines. To take advantage of our partnership with AJE, visit the AJE website (http://aje.com/go/plos) for a 15% discount off AJE services. To take advantage of our partnership with Editage, visit the Editage website (www.editage.com) and enter referral code PLOSEDIT for a 15% discount off Editage services.  If the PLOS editorial team finds any language issues in text that either AJE or Editage has edited, the service provider will re-edit the text for free.

A clean copy of the edited manuscript (uploaded as the new *manuscript* file).

3. "We note that the grant information you provided in the ‘Funding Information’ and ‘Financial Disclosure’ sections do not match. 

Additional Editor Comments:

Dear Dr. Pan,

Your manuscript entitled "Predictive Significance of Joint Plasma Fibrinogen and Urinary Alpha-1 Microglobulin-Creatinine Ratio in Patients with Diabetic Kidney Disease" has been assessed by our reviewers. They have raised a number of points which we believe would improve the manuscript and may allow a revised version to be published in PLOS ONE.

Reviewers' comments:

Reviewer's Responses to Questions

**Comments to the Author**

1. Is the manuscript technically sound, and do the data support the conclusions?

Reviewer #1: Yes

Reviewer #2: Yes

Reviewer #3: Yes

2. Has the statistical analysis been performed appropriately and rigorously? 

Reviewer #1: Yes

Reviewer #2: Yes

Reviewer #3: Yes

3. Have the authors made all data underlying the findings in their manuscript fully available?

Reviewer #1: Yes

Reviewer #2: No

Reviewer #3: Yes

4. Is the manuscript presented in an intelligible fashion and written in standard English?

Reviewer #1: Yes

Reviewer #2: No

Reviewer #3: Yes

5. Review Comments to the Author

Reviewer #1: I read the article with great interest. the authors have investigated the suitability of plasma fibrinogen (FIB) concentration and urinary alpha-1microglobulin-creatinine ( α1-MG/CR) ratio in DKD. The findings in this study are interesting and may have clinical implications. The study concludes that joint plasma FIB concentration and urinary α1-MG/CRratio can be used as a powerful predictor for general and advanced DKD. The study is interesting, the sample size is adequate. I have the following suggestions;

1. The introduction section is not strong, it is poorly written. For this clinical study, a much better introduction should be provided. the importance of early diagnosis of DKD must be emphaszied.

2. The study must present a hypothesis toward the end of the introduction section.

3. I partially agree with the conclusions. While the authors stated that joint plasma FIB concentration and urinary α1-MG/CRratio can be used as a powerful predictor for general and advanced DKD. Looking at the AUC, I believe these values are good for general DKD only, not for advanced disease.

4. How many of the DKD patients were advanced, non-advanced and how many did not have DKD?

5. Discussion of this article can be improved by emphasising on the current knowledge and the potential of novel markers and their impact on healthcare.

Reviewer #2: The study reports on the predictive significance of plasma fibrinogen and urinary alpha1 microglobulin in DKD.

1. Predictive value of plasma fibrinogen in DKD is not a novel finding (ref.12), and urinary alpha1 microglobulin is also an established biomarker of non-specific tubular injury. Therefore, the authors should, as the next step, study a prospective cohort to validate the significance of fibrinogen, for example.

2. Specificity of these markers to DKD should also be investigated by studying non-diabetic CKD patients as a control group.

3. The additional value of combining FIB and a1-MG/CR is also marginal when compared with a1-MG/CR alone, and it is difficult to find advantage of combining these two markers.

Reviewer #3: Summary

This study investigated the predictive significance of joint plasma fibrinogen (FIB) concentration and urinary alpha-1 microglobulin-creatinine (α1-MG/CR) ratio in DKD.. 234 patients with type 2 diabetes were enrolled, including 117 patients with pure T2D and 117 ones with DKD, and their clinical and laboratory data were retrospectively assessed. In 117 DKD patients, no subjects were in stage I, based on the clinical feature of normoalbuminuria, and there were 46 patients in stage II (27 in IIa, 19 in IIb), 35 in stage III, and 36 in stage IV, according to the renal punctures. Results show indicate that plasma FIB and urinary α1-MG/CR levels were higher in patients with DKD than with pure T2D (p<0.001). The multivariate-adjusted odds ratios (ORs) were 5.047 (95%CI: 2.276-10.720) and 36 2.192 (95%CI: 1.539-3.122) (p<0.001) for FIB and α1-MG/CR as continuous variables for DKD prediction, respectively. The optimal cut-off values were 3.21 g/L and 2.11mg/mmol for identifying DKD, and 5.58 g/L and 11.07 mg/mmol for advanced DKD from ROC curves. At these cut-off values, the sensitivity and specificty of joint FIB and α1-MG/CR were 0.95 and 0.92 for indentifying DKD, and 0.62 and 0.67 for indentifying advanced DKD, respectively. The area under curve was 0.972 (95%CI: 0.948-0.995) (p<0.001) and 0.611, 95%CI: 0.488-0.734) (p>0.05). The multivariate-adjusted ORs for joint FIB and α1-MG/CR at the cut-off values were 214.500 (95%CI: 58.054-792.536) and 3.252 (95%CI: 1.040-10.175) (p<0.05), respectively. The conclusion is that joint plasma FIB concentration and urinary α1-MG/CR ratio can be used as a powerful predictor for general and advanced DKD.

Comments

The strength of this study is the confirmation of DKD by renal puncture pathology. However, there are some major concerns to be addressed to support the results and conclusion.

1. The authors divided 117 DKD patients into two groups including test group (59 subjects) and validation group (58 subjects). What is the standard of grouping? Random or others? Is there the possibility of selection bias?

2. Is there DKD patients with normoalbuminuria but elevated SCr levels? What is the result of joint plasma FIB concentration and urinary α1-MG/CR ratio in these patients?

3. The statistical results indicated that joint FIB and α1-MG/CR have a low power in identifying patients with advanced DKD from non-advanced DKD. However, when performing regression analysis by treating FIB and α1-MG/CR as categorical variables based on the optimal cut-off values (5.58 g/L and 11.07 mg/mmol, respectively) from ROC curve analysis, joint FIB and α1-MG/CR demonstrated a high OR value (3.252) in prediction for advanced DKD. How does the author explain this result？

4. In this study, there are some parameters with missing data, the author should present the proportion and processing of missing data.

5. There are typos or spelling mistakes, such as the “parametes” at line 135, page 7. The authors should go through the whole manuscript and make revisions.

---

## [Author Response · Author response to Decision Letter 0]

28 May 2022

Reviewer #1: 

1.The introduction section is not strong, it is poorly written. For this clinical study, a much better introduction should be provided. the importance of early diagnosis of DKD must be emphaszied.

Reply: We added more contents in line 45-55 in Introduction section.

2. The study must present a hypothesis toward the end of the introduction section.

Reply: we have added a hypothesis at the end of the introduction section.

3.I partially agree with the conclusions. While the authors stated that joint plasma FIB concentration and urinary α1-MG/CRratio can be used as a powerful predictor for general and advanced DKD. Looking at the AUC, I believe these values are good for general DKD only, not for advanced disease.

Reply: we have corrected the related discussion in the 2nd, 3rd and 4th paragraphs and the conclusion.

4.How many of the DKD patients were advanced, non-advanced and how many did not have DKD?

Reply: The advanced DKD and non-DKD patients have been defined as stage II+III and stage IV, respectively, in Patients population section. The amounts of advanced DKD and non-DKD were 36 and 81, respectively, which was descripted in Basic characteristics and biomarkers of pure T2D and DKD patients section in the original manuscript. We have revised the description in line 137-138.

5.Discussion of this article can be improved by emphasising on the current knowledge and the potential of novel markers and their impact on healthcare.

Reply: We added some discussion in the second, third and fourth paragraphs of the Discussion section.

Reviewer #2: 

1.Predictive value of plasma fibrinogen in DKD is not a novel finding (ref.12), and urinary alpha1 microglobulin is also an established biomarker of non-specific tubular injury. Therefore, the authors should, as the next step, study a prospective cohort to validate the significance of fibrinogen, for example.

Reply: Our study was a retrospective study, therefore, we aimed to further carry out a prospective study to validate the conclusion. In Conclusions section, we refreshed the statement. 

2.Specificity of these markers to DKD should also be investigated by studying non-diabetic CKD patients as a control group.

Reply: In this study, we did not include non-diabetic CKD, thus, based on the present data, we can not study its specificity. We may include these patients as a control group in further study to draw a more definitive conclusion. 

3.The additional value of combining FIB and a1-MG/CR is also marginal when compared with a1-MG/CR alone, and it is difficult to find advantage of combining these two markers.

Reply: Although a1-MG/CR is a sensitive diagnostic biomarker of tubular injury, and it also exhibited a high identifying value for DKD with an AUC of 0.972, which was similar to that of combining FIB and a1-MG/CR, the combination showed a much higher predictive value than that of a1-MG/CR single use. These results were consistent with the major aim of our study in assessing the predictive significance.

Reviewer #3: 

The strength of this study is the confirmation of DKD by renal puncture pathology. However, there are some major concerns to be addressed to support the results and conclusion.

1.The authors divided 117 DKD patients into two groups including test group (59 subjects) and validation group (58 subjects). What is the standard of grouping? Random or others? Is there the possibility of selection bias?

Reply: We grouped the DKD patients according to the 1:1 principle in random. The grouping method was as follows: we first numbered the patients from No. 1 to 117, then selected 117 numbers from the random number table to assign a random number to each patient. Subsequently, we ranked the 117 random numbers in ascending order, and selected the first 59 cases as the test group, and the last 58 ones as the validation group. Finally, we compared the differences of the variables between the two groups, no significant differences were found (p>0.05 for all), which has been described in Correlation analysis of plasma FIB with urinary proteins section.

2.Is there DKD patients with normoalbuminuria but elevated SCr levels? What is the result of joint plasma FIB concentration and urinary α1-MG/CR ratio in these patients?

Reply: In our original data, there was 6 DKD patients with normoalbuminuria (<3.0 mg/mmoL), but only one patient (male) exhibited elevated sCR level (251.2µmol/L, compared with the median 86.8µmol/L in all T2D patients and the upper limit 133µmol/L of reference interval for males). However, except for the 2 missing results of the combination in the 6 DKD patients, the levels of the rest four were 0.50-1.0, and all were beyond the median of 0.47 in T2D patients. 

3.The statistical results indicated that joint FIB and α1-MG/CR have a low power in identifying patients with advanced DKD from non-advanced DKD. However, when performing regression analysis by treating FIB and α1-MG/CR as categorical variables based on the optimal cut-off values (5.58 g/L and 11.07 mg/mmol, respectively) from ROC curve analysis, joint FIB and α1-MG/CR demonstrated a high OR value (3.252) in prediction for advanced DKD. How does the author explain this result？

Reply: Within our knowledge, we think that the ROC analysis can indicate the diagnostic power, but the regression analysis can reveal the predictive power. Therefore, joint FIB and α1-MG/CR may be have a relatively lower identifying value and a higher predictive significance for advanced DKD. In fact, the result of univariate analysis of the combination did not show an OR with statistical difference, and the levels also exhibited no statistical differenc between the three stages. Based on these results, we revised the text in the fourth paragraph of the Discussion section and in Conclusion section.

4. In this study, there are some parameters with missing data, the author should present the proportion and processing of missing data.

Reply: We presented the percent of missing data and the processing method in ;ine 144-148 of the Basic characteristics and biomarkers of pure T2D and DKD patients section.

5. There are typos or spelling mistakes, such as the “parametes” at line 135, page 7. The authors should go through the whole manuscript and make revisions.

Reply: We corrected the mistakes we could found.

---

## [Editor Report · Decision Letter 1]

27 Jun 2022

Predictive significance of joint plasma fibrinogen and urinary alpha-1 microglobulin- creatinine ratio in patients with diabetic kidney disease

PONE-D-22-06480R1

Dear Dr. Zhu,

We’re pleased to inform you that your manuscript has been judged scientifically suitable for publication and will be formally accepted for publication once it meets all outstanding technical requirements.

Kind regards,

Donovan Anthony McGrowder, PhD., MA., MSc

Academic Editor

PLOS ONE

Additional Editor Comments (optional):

Dear Dr. Zhu,

The manuscript was revised in accordance with the reviewers’ comments and is provisionally accepted pending final checks for formatting and technical requirements.

Regards,

Dr. Donovan McGrowder (Academic Editor)<o:p></o:p>

---

## [Editor Report · Acceptance letter]

30 Jun 2022

PONE-D-22-06480R1 

Predictive significance of joint plasma fibrinogen and urinary alpha-1 microglobulin-creatinine ratio in patients with diabetic kidney disease 

Dear Dr. Zhu:

I'm pleased to inform you that your manuscript has been deemed suitable for publication in PLOS ONE. Congratulations! Your manuscript is now with our production department. 

Kind regards, 

on behalf of

Dr. Donovan Anthony McGrowder 

Academic Editor

PLOS ONE